# Role of Dairy Foods, Fish, White Meat, and Eggs in the Prevention of Colorectal Cancer: A Systematic Review of Observational Studies in 2018–2022

**DOI:** 10.3390/nu14163430

**Published:** 2022-08-21

**Authors:** Iker Alegria-Lertxundi, Luis Bujanda, Marta Arroyo-Izaga

**Affiliations:** 1Department of Pharmacy and Food Sciences, Faculty of Pharmacy, University of the Basque Country UPV/EHU, 01006 Vitoria-Gasteiz, Spain; 2Department of Liver and Gastrointestinal Diseases, Biodonostia Health Research Institute, 20014 San Sebastian, Spain; 3Department of Gastroenterology, University of the Basque Country UPV/EHU, 20018 San Sebastian, Spain; 4Consortium for Biomedical Research in Hepatic and Digestive Diseases, CIBERehd, 28029 Madrid, Spain; 5BIOMICS Research Group, Microfluidics & BIOMICS Cluster, University of the Basque Country UPV/EHU, 01006 Vitoria-Gasteiz, Spain

**Keywords:** adults, case–control studies, colorectal cancer, dairy, eggs, fish, poultry, cohort studies, systematic review, white meat

## Abstract

There is limited evidence to support the relationship between the consumption of animal-source foods other than red meat and processed meat and colorectal cancer (CRC) risk. We aimed to examine the recent available evidence from observational studies about the association between these food groups’ intake and CRC risk. For this systematic review, we searched the PubMed database for the last five years. A total of fourteen cohort studies and seven case–control studies comprising a total of >60,000 cases were included. The studies showed a consistent significant decrease in CRC risk, overall and by subsites, associated with a high consumption of total dairy products. Less strong effects associated with the consumption of any subtype of dairy product were observed. Fish consumption, overall and by subtypes (oily or non-oily and fresh or canned), showed a mild inverse association with CRC risk. The association between white meat and egg intake and CRC risk was low and based on a small number of studies; thus, these findings should be interpreted with caution. In conclusion, a high consumption of total dairy products was associated with a lower CRC risk. However, evidence for fish, white meat, and eggs and the CRC risk were not as strong.

## 1. Introduction

Colorectal cancer (CRC) is the third-most frequent cancer and the second-highest mortality in cancer patients worldwide [1]. The global cancer statistics in 2020 showed there were about 1932 million new cases and 935,000 deaths of CRC worldwide, accounting for 10.0% of the total new cases of cancer and 9.4% of the total cancer-related deaths, respectively [1].

During recent years, the mortality rates for CRC have been decreasing due to early screening programs [2,3] and better treatment options [4]. However, the aetiology of CRC is complex and still not fully understood. Both genetic and environmental factors play an important role in the aetiology of this disease [5]. Environmental and, in particular, diet and lifestyle factors are likely to be the main determinants of CRC development [6].

There is considerable evidence to suggest that the consumption of a diet with high intakes of vegetables, fruits, and whole grains may decrease the risk of CRC and that the consumption of processed meat and alcohol are risk factors for this type of cancer [7]. In addition, the Mediterranean dietary pattern could reduce the overall cancer risk [8] and, in particular, CRC risk [9]. On the other hand, recently, CRC innovative treatments deriving from natural extracts, termed nutraceuticals, have gained interest [10]. In this sense, for example, anti-inflammatory and reparative properties have been attributed to the nutraceutical grape seed extract [11].

However, evidence for the consumption of animal-source foods other than red meat and processed meat, as is the case with dairy products, fish, white meat, and eggs, is not as strong [7]. The latest report from the Continuous Update Project (CUP), led by the World Cancer Research Fund/American Institute for Cancer Research (WCRF/AICR), concluded that there is strong evidence that consuming dairy products decreases the risk of cancer [7]. However, the evidence suggesting that the consumption of different types of dairy products (i.e., milk, yogurt, cheese or subtypes by fat content) is limited [12]. Regarding fish consumption, the evidence suggesting that the consumption of fish decreases the risk of CRC is limited but generally consistent [7], although the risk of CRC associated with the consumption of different types of fish (oily or non-oily fish) remains unclear [13]. In addition, to date, studies on the possible association between white meat (such as poultry—chicken, turkey, duck, and goose—and rabbit) or eggs and CRC risk are scarce, and the results are unclear [14]. 

Even if the most current dietary guidelines advocate the consumption of moderate intakes of low-fat dairy foods, fish, white meats, and eggs, in the context of a healthy diet to prevent chronic disease [15,16], the American Cancer Society does not issue specific recommendations on the consumption of these food groups for cancer prevention. Therefore, further research on the association between the consumption of these foods and CRC risk should be of considerable interest in terms of public health. A better understanding of the impact of these categories of animal-source foods would be of great help to make recommendations regarding these products.

This systematic review aims to examine the recent available evidence from observational studies (cohorts and case–control) in adults about the association between the consumption of animal-source foods other than red meat and processed meat, as is the case with dairy products, fish, white meat, and eggs, and the CRC risk. We also investigated associations with specific types of these food groups (e.g., whole-fat dairy products or canned fish, and whether the associations with CRC risk depended on the CRC subsite (colon or rectal and colon cancer location (proximal or distal colon)). The reason the review was based on the publications during the last five years rather than the entire past is that the latest report from the CUP about diet, nutrition, physical activity, and CRC was revised in 2018 [7].

## 2. Methods

### 2.1. Search Strategy

A systematic search was conducted in the PubMed database from the last five years (from 1 January 2018 to 15 July 2022) for published case–control or cohort studies evaluating the associations between the consumption of dairy products, fish, white meat, or eggs and CRC (total CRC, colon cancer (CC), or rectal cancer (RC) and proximal or distal colon cancer). For this purpose, a search for the relevant keywords and medical subject heading terms related to the consumption of the abovementioned food groups and subtypes in combination with the keywords related to CRC events was conducted. The search was expanded through citation chaining (forward and backward) of the included studies. Reference lists of all the identified articles and other related review articles, systematic reviews, and meta-analyses were hand-searched for additional articles. The present search was developed according to the “PRISMA Statement” guidelines (see the PRISMA checklist in the Appendix A) (www.prisma-statement.org, accessed on 3 May 2022). For this review, a protocol was not prepared or registered.

Next, the search strategy is detailed:



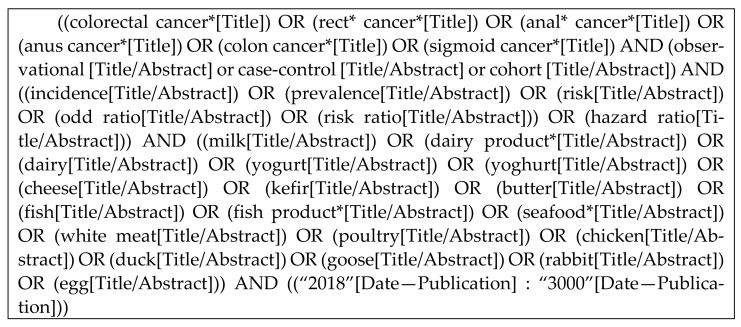



### 2.2. Review Process and Selection Criteria

Two researchers independently screened the titles and abstracts of the articles to identify potentially relevant studies. Studies that passed the title/abstract review were retrieved for a full-text review. An article was retained for the analysis only if: (i) it was published between 2018 and 2022; (ii) it was conducted on humans (>18 y old); (iii) it was written in English or Spanish; (iv) it was an observational study (i.e., case–control or cohort); (v) the exposure variable was at least one of the following dietary components: milk, dairy products, fish, white meat, or eggs; (vi) colorectal adenocarcinoma incidence was the outcome (studies investigating benign adenomas or polyps were excluded); (vii) provision of adjusted odds ratios (ORs) or risk ratios (RRs) or hazard ratios (HRs) with 95% confidence intervals (CIs); (viii) the estimates were adjusted for age; and (ix) Newcastle–Ottawa scale (NOS) ≥ 4, indicating sufficient study quality [17]. Template data collection forms and data extracted from the included studies will be made available upon request from the corresponding author.

The following types of publications were excluded: (i) nonoriginal papers (e.g., reviews, meta-analyses, book chapters, commentaries, editorials, proceedings, or letters to the editor); (ii) ecologic assessments and correlation studies; (iii) non-peer-reviewed articles; (iv) off-topic studies (e.g., those who analysed dietary patterns); (v) studies on CRC mortality; (vi) studies lacking specific CRC data; (vii) animal and mechanistic studies; (viii) studies conducted at stages of life other than adulthood (e.g., childhood and adolescence); (ix) supplements to the main manuscript; (x) duplicate publications; and (xi) those with a NOS score < 4.

### 2.3. Study Quality Assessment

To evaluate the validity of the individual studies, two reviewers worked independently to determine the quality of the included studies based on the use of the NOS for case–control or cohort studies [17]. The maximum score was 9, and a high score (≥6) indicated a high methodologic quality; however, given the lack of studies on the subject under study, it was agreed to select those that had a score equal to or greater than 4. A consensus was reached between the reviewers if there were any discrepancies.

### 2.4. Data Extraction

The data extracted for each individual study included the following: study design; name of the first author; year of publication; characteristics of the study population (age range or mean age, sex, and country); dietary exposure; dietary assessment instrument used; outcomes (including cancer site); OR; RR or HR (95% CI); adjusted variables; NOS; and funding sources. For case–control studies, the following additional information was extracted: number of cases and number of controls. For cohort studies, the following additional information was extracted: number of participants at the baseline, number of diagnosed cases of CRC, and length of follow-up. The most relevant variables to the outcome studied, thus, when the researchers used multiple estimates, the one with the highest number of adjusted variables was recorded. Where multiple estimates for the association of the same outcome were used, the one with the highest number of adjusted variables was extracted.

## 3. Results and Discussion

### 3.1. Study Selection

Figure 1 shows the PRISMA flow diagram summarizing the identification and selection of the relevant publications. Twenty-one studies were included in the systematic review: fourteen cohort studies [18,19,20,21,22,23,24,25,26,27,28,29,30,31] and seven case–control studies [32,33,34,35,36,37,38].

### 3.2. Study Characteristics about Milk and Dairy Products

Table 1 and Table 2 show the main characteristics of the studies selected about the association between the consumption of dairy products and CRC risk. In total, the cohort studies included 1,847,899 participants with 18,880 cases recorded during follow-up periods that ranged from 6 to 32 y. The case–control studies included 3553 cases and 5902 controls. Table 3 summarizes the association results obtained in the articles reviewed.

Of the cohort studies, six were conducted in Europe (one in various European countries, one in France, one in Norway, one in Spain, and two in Sweden); one in China; and two in the United States. The case–control studies were conducted in three countries (China, Iran, and Spain). Most of the studies obtained funding only from agencies (in particular, nine of eleven studies), and in the rest, no information was provided on funding resources or grants.

### 3.3. Dairy Products

#### 3.3.1. Total Dairy Products in Overall and by Fat Content

Five cohort study [19,20,21,26,27] and three case–control study [32,33,34] comparisons were used to assess the association between the total dairy consumption and CRC risk. In three of the cohort studies [19,20,21], a significant inverse association was observed for CRC, in addition to CC and RC separately and, also, in subtypes according to tumour localisation (distal and proximal). Previous studies have shown a significant decrease in the risk of CRC risk associated with a high consumption of total dairy products [12], supporting the conclusions of the last systematic review and meta-analysis of prospective studies, which updated the evidence of the WCRF-AICR Continuous Update Project [39]. Regarding the total dairy product consumption and CRC risk, the aforementioned meta-analysis observed similar associations in men and women. This result agrees with the study of Zhang et al. [33] included in the present search.

Observed inverse associations between intake of dairy products and CRC development were mostly attributed to their high calcium content. Additionally, the casein and lactose may increase calcium bioavailability, and lactic acid-producing bacteria may also protect against CRC. Other nutrients or bioactive compounds present in dairy products, such as lactoferrin, the short-chain fatty acid butyrate, or vitamin D (from fortified products), could also be protective factors for this type of cancer; however, further studies are required to confirm this effect [40].

However, in the other two cohort investigations [26,27] and one of the case–control researches [34] analysed in the present review, CRC did not show clear associations with the total dairy consumption. Moreover, in one case–control study [32], a direct association was found between total dairy intake and CRC risk. The discrepancies between the findings of these studies could be due to differences in the amount and types of products consumed, as well as differences in the overall lifestyle between the population studied (for example, in China, generally, the levels of dairy consumption are low, but people who have a high consumption of dairy products can also have a high intake of red and processed meat, due to the Westernisation of eating habits). Moreover, these discrepancies could be due to differences in the adjustment variables included in the statistical models.

Regarding the associations between dairy products by fat content and the risk of CRC, only one of the analysed studies [19] researched this aspect and concluded that whole-fat dairy consumption was not associated with an increased CRC risk. This finding agrees with those previously reported [41,42] and allows us to deduce that there are no reasons to not recommend whole-fat dairy consumption. The associations between the consumption of dairy products with different fat contents and CRC risk were not documented in the latest report by the CUP panel [7]. In a recent meta-analysis, it was observed that a high consumption of high-fat dairy products was a significant inverse association with the CRC risk [12]. However, due to the substantial heterogeneity among the few studies analysed in the aforementioned meta-analysis, Barrubés et al. [12] concluded that these observations should be interpreted cautiously. 

#### 3.3.2. Total Milk, Whole, and Low-Fat Milk

The analysis of the association of the total milk consumption with CRC risk included six cohort study [18,19,22,24,25,26] and two case–control study [33,34] comparisons. No significant association was found in any of them, except for a weak inverse association with CC risk in one of the cohort studies [18] and a significant inverse association with the CRC risk in one of the case–control studies [33] and one of the cohort studies [25], this last association independent of sex and cancer sites [25,33]. Previously, cohort studies on the association between milk intake and the risk of CRC have reported different results [43,44,45,46,47,48]; in some of them, an inverse association between milk intake and CC risk was also found, in particular among women in an EPIC study [41]. In this sense, it should be noted that the aetiological factors may differ between CC and RC [49]; in fact, different genetic features between CC and RC have been observed [50]. In addition, milk may have different effects on CC and RC risk [51].

Continuing with the study of milk consumption, but in this case, depending on the fat content, one of the analysed studies [19] found suggestive evidence that a high consumption of low-fat milk is associated with lower CRC incidence, this beverage as the main contributor to total dairy product consumption in the studied population. Barrubés et al. [19] explained that they found no significant association between the whole-fat milk intake and CRC risk, due probably to the low consumption of this subtype of milk in the population studied. In any case, the fat content in whole-fat milk might mitigate the potential benefits of the other bioactive components [52].

#### 3.3.3. Yogurt and Other Fermented Dairy Products

Concerning fermented dairy products, five cohort studies [19,23,24,25,26], and one case–control study [34] were included in this review of the association between the consumption of these products and CRC risk. In four of these five cohort investigations [19,24,25,26], no significant associations were found for fermented dairy products or for yogurt and different types of yogurts (low-fat and whole-fat yogurt).

Only in one of the analysed cohort studies [23], yogurt consumption was associated with a reduced risk of proximal CC with a long latency period, which confirmed the results from the previously conducted European Prospective Investigation into Cancer and Nutrition (EPIC). Similar results were found in a previous analysis [53] and a pooled analysis of prospective cohorts [54]. In this same vein, one case–control study analysed did find a significant inverse association between yogurt consumption and proximal CC, with a dose–response relationship [34]. This result agrees with that of previous case–control investigations—in particular, with one study in Los Angeles in which the authors observed an inverse association between regular yogurt consumption and CC [55] and another in the Italian EPIC population after controlling for the calcium and other nutrients intake [56]. However, no relation was evident in either the PREvention with Mediterranean Diet Study [19] or the EPIC after adjustment for dietary calcium [41]. Possible explanations for the diverse findings are among other differences in the study designs, dietary assessment methods, genetic backgrounds, lifestyle, and dietary habits of study populations. The consumption of yogurt varies greatly worldwide, both the consumption of any type of yogurt and plain yogurt without added sugar, fruit, or flavourings is more frequent in Europe than in the United States [57,58,59,60]. Moreover, specifically, the findings related to distal CC could be due, at least in part, to the confounding effect of calcium intake.

#### 3.3.4. Cheese

Five cohort [19,22,24,25,26] and two case–control studies [28,30] were used to analyse the associations between the consumption of cheese and CRC risk. These cohort studies did not support any major adverse or beneficial effects of cheese in the diet from the CRC risk perspective, which agrees with the conclusions of the WCRF/AICR [41], which indicates the association with cheese consumption was not clear. In any case, two results found in the analysed cohort studies should be highlighted. First, a high cheese consumption was associated with a modestly decreased risk of CRC [24,25], which has been attributed to positive effects of lactoferrin [61] and dairy lipids [62], as well as changes in the gut microbiome [63]. Second, the consumption of “fromage blanc” (a French type of quark/cottage cheese resulting from lactic coagulation and draining without further processing or additives) was associated with an increased risk of CRC [26]. The authors did not find any clear mechanism to explain the observed association with the consumption of “fromage blanc”, so they suggested that their results be interpreted with caution, because it could result from an artifact of the study sample. On the other hand, in one of the case–control studies [32], a direct association was found for high-fat cheese consumption, which was explained by the saturated fat content and by the increased bile acid discharge. It has been reported that an increase in bile acid production above the physiological levels promotes CRC [64,65].

#### 3.3.5. Other Dairy Products: Butter, Sugary Dairy Products, Cream, and Ice Cream

Three cohort studies [19,24,26] were used to compare the overall risk of CRC between the highest and lowest consumption of butter and sugary dairy products. As for the butter, no significant associations were found in two of these manuscripts [19,24]. Regarding sugary dairy products, Barrubés et al. [19] did not find an association with the CRC risk. However, Deschasaux-Tanguy et al. [26] observed a direct association with CRC risk. These authors argued that the observed association could be due to the fact that these products often contain elevated amounts of sugar, as well as additives (for instance, emulsifiers or texturizers) [66], which could increase the CRC risk through body weight gain and, consequently, increase the insulin resistance [7].

Regarding cream and ice cream intake, only in one of the case–control studies analysed were they assessed [34]. Collatuzzo et al. [34] found a direct association for cream and CRC in the overall CC, as well as proximal CC, while ice cream was inversely associated with the risk of distal CC. In relation to the first of these associations, researchers pointed out that the literature reports a direct association between high-fat dairy product intake and overall cancer mortality and between a low-fat dairy intake and cancer mortality but no relation with CRC in particular [19]. As mentioned above in the subsection on the total dairy products by the fat content, there was no evidence of an increased CRC risk derived from whole-fat dairy consumption [19].

### 3.4. Study Characteristics about Fish, White Meat, and Eggs

Table 4 and Table 5 show the main characteristics of the studies selected about the association between the consumption of fish, white meat, and eggs and CRC risk. In total, the cohort studies included 1,438,288 participants with 38,508 cases recorded during follow-up periods that ranged from 5.7 to 30 y. The case–control studies included 5203 cases and 8430 controls. Table 6 summarises the associated results obtained in the articles reviewed. Of the cohort studies, two were conducted in Europe (one in various European countries and one in Denmark), two in the UK, and one in the United States. The case–control studies were conducted in Asia (one in China and one in the Republic of Korea), Europe (one in Italy and one in Spain), and in Morocco. Most of the studies obtained funding only from agencies (in particular, nine out of ten studies), and in one, no information was provided on the funding resources or grants.

### 3.5. Fish, White Meat, and Eggs

#### 3.5.1. Fish

Three cohort [22,28,31] and three case–control studies [32,36,38] were used to analyse the associations between the highest and lowest consumption of total fish and CRC risk. In two of these cohort investigations [28,31] and in one of these case–control studies [38], fish consumption was associated with a significant CRC risk reduction. A protective effect was also observed for oily fish in both the cohort [28,31] and case–control studies [32]. Aglago et al. [28] found an inverse association between oily fish and non-oily fish consumption, separately, both for the overall CRC risk, as well as for CC and distal CC alone. Moreover, in the cohort study [31] and the case–control study [38] that analysed the consumption of canned fish, the results were significant and showed an inverse association between said intake and CRC risk. 

These findings agree with those previously reported in two recent meta-analyses [67,68]. In the meta-analysis of Caini et al. [67], it is suggested that the aforementioned association may be attributable to several biological mechanisms, such as those related to ω-3 polyunsaturated fatty acids (PUFAs) that: (i) affect eicosanoids metabolism [69], (ii) are incorporated into membrane phospholipids, and (iii) that do not enhance the luminal concentration of secondary bile acids and the lower colon and liver activity of ornithine decarboxylase and tyrosine-specific protein kinase, all these mechanisms involved in colon carcinogenesis [70]. 

In addition, ω-3 PUFA has been associated with a higher intestinal microbial diversity, thus improving the host immune function and eventually decreasing the risk of the development of CRC [71,72]. Mechanisms related to eicosapentaenoic acid and docosahexaenoic acid that produce lipid mediators endowed with pro-resolving, immunomodulatory, and anti-inflammatory properties could also be involved [73]. On the other hand, the association between fish consumption and CRC risk could also be due, in part, to: (i) a replacement effect, since those who eat more fish generally eat less red meat, whose causal link with CRC is well-known, and (ii) the fact that preferring fish instead of meat may be part of a generally healthier lifestyle, including protective habits against this type of cancer [74,75].

#### 3.5.2. White Meat

Five cohort studies [22,29,30,31,37] and one case–control study [35] were used to analyse the association between high and low white meat intake and CRC risk. In one of the cohort studies [29], poultry consumption was inversely associated with RC risk in women but positively in men. The authors argued that these differences by sex could be due to a lower incidence of this type of cancer among women and also to a less frequent consumption of meat among women. In the case–control study [35], a positive association was observed between the poultry intake and CC risk among men. In contrast, no significant association was observed between white meat and CRC risk among women. The same as Knuppel et al. [29], Deoula et al. [35] considered that their results may be due to a lower consumption of the total meat among women compared to men.

#### 3.5.3. Eggs

Two case–control studies [32,37] were used to analyse the association between the high and low consumption of eggs and CRC risk. In one of them, an inverse association was found, and in the other, no association was found. Although Shen et al. [37] did not provide arguments to explain why egg consumption is a protective factor against CRC risk, it should be noted that eggs contain several antioxidants (including selenium, carotenoids, and vitamin E), which reduce the free radicals arising from cellular metabolism [76]. Positive associations between oxidative stress and the incidence of chronic diseases such as cancers have been reported [77]. In addition, eggs are a good source of choline and B vitamins directly or tangentially involved in one-carbon metabolism [76]. Disorders in one-carbon metabolism can lead to decreased DNA synthesis, genomic instability, and decreased methyl donor production [78]. Genomic instability and DNA hypomethylation are common traits of CRC [79]. On the contrary, in the meta-analysis of Schwingshackl et al. [14], based on a small number of studies (*n* = 3), the results showed a positive significant association in the high vs. low meta-analysis. A plausible hypothesis to explain these observations is that a high cholesterol intake would increase the formation of secondary bile acid and enhance the induction of colorectal tumours [80].

### 3.6. Strengths and Limitations

This systematic review has several strengths: (i) cohort and case–control studies through a systematic search have been identified, (ii) a quantitative NOS scale was used to evaluate the quality of the individual studies, and (iii) all of the studies used a validated questionnaire to assess the food consumption. 

The limitations of this review include: (i) the heterogeneous nature of the studies, including the study population characteristics, sample size, study design, and follow-up periods; (ii) being observational studies, residual confounding can be a problem, and not all studies were adjusted for important confounders; (iii) in the same way, since these are observational studies, the results cannot support the causal relationship between food group consumption and CRC risk; and (iv) the fact that some studies used self-reported data on the dietary intake may affect the reliability of the reported data, although the use of validated questionnaires could reduce this bias.

## 4. Conclusions

This systematic review of observational studies supports the protective role of the consumption of total dairy products against the CRC risk in all CRC subsites. Therefore, it makes sense to suggest that the dairy intake might be associated with a lower risk of CRC and, furthermore, that there are no reasons to advise against whole-fat dairy products. However, the effect associated with the consumption of any subtype of dairy product, including milk, fermented dairy products (including yogurt), cheese, and other dairy products, was less strong. As regards fish consumption, a mild inverse association with CRC risk was observed, with a similar effect for CC and RC, according to the tumour location. This same relationship was found for different types of fish (oily and non-oily fish), both fresh and processed (canned fish). Finally, this review on white meat and eggs was based on a small number of studies, and the evidence was low; thus, the findings for these food groups should be interpreted with caution. The current findings related to dairy products and fish confirm those from previous meta-analyses [7,12,67,68]. The main new findings of the present review are those related to specific subtypes of fish (in particular, canned fish), white meat, and eggs. Further studies are needed to confirm these results and to clarify the mechanism to explain the observed associations, with special emphasis on the differences in the subtypes of foods and CRC subsite-specific risk.

## Figures and Tables

**Figure 1 nutrients-14-03430-f001:**
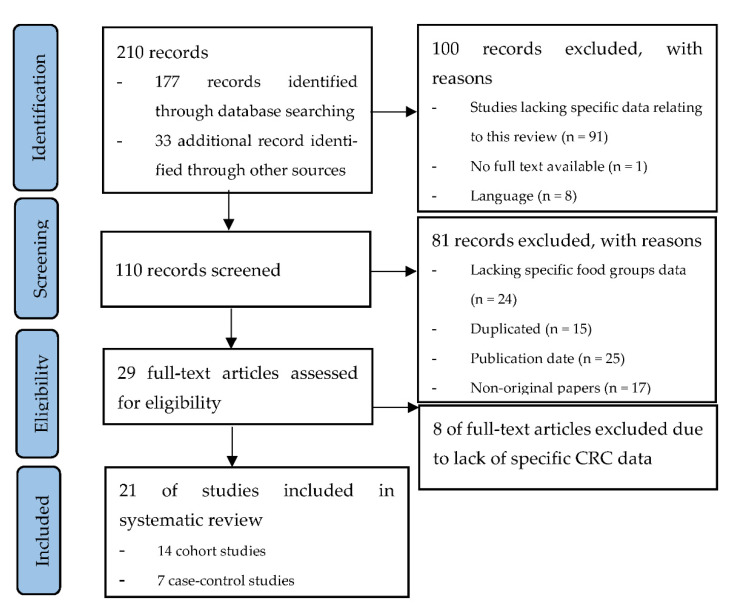
PRISMA flow diagram summarizing the identification and selection of the relevant publications about the associations between the consumption of dairy products, fish, white meat, and eggs and the risk of colorectal cancer.

**Table 1 nutrients-14-03430-t001:** Characteristics of the ten cohort studies included in the systematic review examining the association between the consumption of dairy products and the risk of colorectal cancer.

Study, Year (Ref.)	Study Cohort, Country (Age, y)	No. Participants (M/W)	No. Incident Cases	Follow-Up Length, y	Exposure	HR (95%CI)	Adjustments to HR	NOS Quality Score
Bakken et al., 2018 [18]	Norway: NOWAC Cohort Study (median, 51)	81,675 W	872 CRC (617 CC, 255 RC)	6	Total milk: >240 g/day vs. never/seldom	CRC: 0.85 (0.69, 1.05) CC: 0.80 (0.62, 1.03) RC: 0.97 (0.67, 1.42)	Age as the time scale and adjusted for BMI, smoking, processed meat, red meat, hard white cheese, yogurt, fibre from foods, alcohol, and energy intake	6
Barrubés et al., 2018 [19]	Spain: PREDIMED trial (55–80)	7216 M&W	97 CRC	6	Total dairy products: 564 g/day vs. 206 g/day	CRC: 0.55 (0.31, 0.99)	Stratified by recruitment centre. Adjusted for the intervention group, sex, age, leisure time PA, smoking status, family history of cancer, education level, history of diabetes, use of aspirin treatment, and cumulative average consumption of vegetables, fruits, legumes, cereals, fish, meat, olive oil and nuts, and alcohol	7
					Whole-fat dairy products: 114 g/day vs. 0 g/day	CRC: 1.01 (0.62, 1.64)		
					Low-fat dairy products: 495 g/day vs. 67 g/day	CRC: 0.62 (0.36, 1,07)		
					Total yogurt: 128 g/day vs. 8 g/day	CRC: 0.94 (0.56, 1.59)		
					Low-fat yogurt: 122 g/day vs. 1 g/day	CRC: 1.06 (0.65, 1.73)		
					Whole-fat yogurt. 45 g/day vs. 0 g/day	CRC: 0.86 (0.51, 1.46)		
					Cheese: 44 g/day vs. 11 g/day	CRC: 1.23 (0.74, 2.06)		
					Total milk: 449 g/day vs. 117 g/day	CRC: 0.63 (0.36, 1.10)		
					Low-fat milk: 407 g/day vs. 15 g/day	CRC: 0.54 (0.32, 0.92)		
					Whole milk: 60 g/day vs. 0 g/day	CRC: 1.06 (0.64, 1.75)		
					Concentrated full-fat dairy products: 45 g/day vs. 11 g/day	CRC: 1.11 (0.66, 1.86)		
					Sugar-enriched dairy products: 14 g/day vs. 0 g/day	CRC: 0.98 (0.55, 1.75)		
					Fermented dairy products: 166 g/day vs. 36 g/day	CRC: 0.90 (0.53, 1.53)		
Vulcan et al., 2018 [20]	Sweden: Malmö Diet and Cancer Cohort Study (cases, 60,6+/−7,0; non-cases, 58,0+/−7,6)	10,966/16,955	923 CRC (590 CC, 317 RC, 16 SCRC)	18	Total dairy products: Q_5_ vs. Q_1_	CRC: 0.77 (0.62, 0.96) CC: 0.81 (0.61, 1.06) RC: 0.66 (0.46, 0.94)	Sex, age, method version, season, total energy, education, PA, and BMI	6
Um et al., 2019 [21]	USA: Iowa Women’s Health Study (55–69)	35,221 W	1731 CRC (971 PCC, 760 DCC)	26	Total dairy products: Q_5_ vs. Q_1_	PCC: 0.87 (0.69, 1.10) DCC: 0.69 (0.53, 0.90)	Age, family history of CRC, BMI, smoking, alcohol, PA, HRT use (W), total energy intake, vitamin D, magnesium, fruit and vegetable intake, red and processed meat intake, dietary oxidative balance score, and supplemental calcium	7
Bradbury et al., 2020 [22]	UK: UK Biobank Cohort Study (40–69)	219,329/256,252	2609 CRC	5.7	Milk: ≥300 mL/day vs. never	CRC: 0.93 (0.87, 1.01)	Stratified by age, sex, geographical region, and socio-economic status. Adjusted for education, smoking status, waist circumference, height, PA, alcohol intake, family history of CRC, aspirin or ibuprofen use, use of vitamin D and folate supplements, and for W: parity, menopause, OCA, and HRT use	6
					Cheese: ≥5 times/week vs. <once/week	CRC: 1.09 (0.96, 1.23)		
Michels et al. 2020 [23]	USA: NHS and HPFS (mean at baseline: M 52.3 and W 45.7)	43,269/83,054	2666 CCR * (1965 CC, 579 RC)	26 M, 32 W	Yogurt: never or <1 serving/mo vs. >1 servings/week	CRC: 0.89 (0.80, 1.00) CC: 0.87 (0.76, 0.99) PCC: 0.84 (0.70, 0.99) DCC: 0.91 (0.74, 1.12) RC: 0.95 (0.76, 1.21)	Age, 2-year follow-up cycle, family history of CRC, history of lower gastrointestinal endoscopy, BMI, height, PA, pack-years of smoking before age 30, current multivitamin use, regular aspirin or NSAIDs use, total caloric intake, alcohol consumption, and energy-adjusted intakes of folate, calcium, vitamin D, total fibre, unprocessed red meat, and processed meat, and for W: parity and age at first birth, menopausal status, age at menopause and HRT	6
Nilsson et al., 2020 [24]	Sweden: VIP and MONICA (25–75)	53,157/52,734	1381 CRC	20	Non-fermented milk: Q_5_ vs. Q_1_	CRC (M): 0.87 (0.67, 1.14) CRC (W): 0.88 (0.68, 1.14)	Age, screening year, dairy product category, BMI, civil status, education level, PA in leisure time, smoking status, recruitment cohort (VIP or MONICA), and Qs of fruit and vegetables, alcohol, and energy intake	8
					Fermented milk: Q_5_ vs. Q_1_	CRC (M): 0.98 (0.77, 1.25) CRC (W): 0.90 (0.70, 1.15)		
					Butter: Q_5_ vs. Q_1_	CRC (M): 0.99 (0.76, 1.28) CRC (W): 0.82 (0.62, 1.08)		
					Cheese: Q_5_ vs. Q_1_	CRC (M): 0.86 (0.67, 1.10) CRC (W): 0.82 (0.63, 1.07)		
Papadimitriou et al., 2021 [25]	Europe: EPIC (35–70)	112,170/274,622	5069 CRC	14.1	Milk (standardized continuous variable)	CRC: 0.96 (0.93, 0.99)	Total energy intake, smoking status, BMI, PA, diabetes history, education status, age sex, and recruitment centre	8
					Cheese (standardized continuous variable)	CRC: 0.95 (0.92, 0.99)		
					Yogurt (standardized continuous variable)	CRC: 0.98 (0.95, 1.01)		
Deschasaux-Tanguy et al. 2022 [26]	France: NutriNet-Santé Cohort Study (42.2+/−14.5)	21,572/79,707	182 CRC	10	Total dairy products: continuous per 1 serving increment	CRC: 1.05 (0.93, 1.19)	Age, sex, height, BMI, baseline type 2 diabetes, prevalent hypertriglyceridemia, hypercholesterolemia, energy intake without alcohol, sugar intake, sodium intake, fruits and vegetables intake, whole foods, red and processed meat consumption, non-dairy calcium intake, non-dairy SFA intake, alcohol intake, number of 24 h dietary records, smoking status, educational level, PA and family history of cancer	6
					Milk: continuous per 1 serving increment	CRC: 0.92 (0.74, 1.15)		
					Yogurt: continuous per 1 serving increment	CRC: 0.90 (0.67, 1.19)		
					Cheese: continuous per 1 serving increment	CRC: 1.10 (0.9, 1.30)		
					*Fromage blanc*: continuous per 1 serving increment	CRC: 1.39 (1.09, 1.77)		
					Fermented dairy products: continuous per 1 serving increment	CRC: 1.10 (0.96, 1.27)		
					Sugary dairy dessert: continuous per 1 serving increment	CRC: 1.58 (1.01, 2.46)		
Kakkoura et al. 2022 [27]	China: China Kadoorie Biobank Study (35–74)	205,000/295,000	3350 CRC	10.8	Total dairy products: never/rarely intake	CRC: 1.00 (0.94, 1.06)	Stratified by age-at-risk, sex, and individual regions. Adjusted for education, income, smoking, alcohol consumption, total PA, family history of cancer, fresh fruit consumption, soy consumption, and BMI	8
					Monthly intake	CRC: 1.10 (1.00, 1.21)		
					Regular intake	CRC: 1.09 (1.01, 1.18)		
					Per 50 g/day of usual intake	CRC: 1.08 (1.00, 1.17)		

BMI: body mass index; CC: colon cancer; CI: confidence interval; CRC: colorectal cancer; DCC: distal colon cancer; EPIC: European Prospective Investigation into Cancer and Nutrition; HPFS: Health Professionals Follow-Up Study; HR: hazard ratio; HRT: hormone replacement therapy; M: men; mo: moth; MONICA: Northern Sweden Monitoring of Trends and Determinants in Cardiovascular disease; NHS: Nurses’ Health Study; NSAIDs: non-steroidal anti-inflammatory drugs; NOS: Newcastle-Ottawa Scale; NOWAC: Norwegian Women and Cancer; OCA: oral contraceptive agent; PA: physical activity; PCC: proximal colon cancer; PREDIMED: PREvención con DIeta MEDiterránea Study; Q: quintile; RC: rectal cancer; SCRC: synchronous colon and rectal cancer; SFA: saturated fatty acid; VIP: Västerbotten Intervention Programme; W: women. * One hundred and twenty-two cases with unknown sites within the colorectum.

**Table 2 nutrients-14-03430-t002:** Characteristics of the three case–control studies included in the systematic review examining the association between the consumption of dairy products and the risk of colorectal cancer.

Study, Year (Ref.)	Country (Age, y)	No. Cases and Endpoint	Sex, No. of Cases (M/W)	No. Controls and Type	Exposure	OR (95% CI)	Adjustments to OR	NOS Quality Score
Alegria-Lertxundi et al., 2020 [32]	Spain (50–69)	308 CRC (74 PCC, 234 DCC)	204/104	308 C	Milk/dairy products: T_3_ vs. T_1_	CRC: 1.80 (0.95, 3.42)	Age, sex, BMI, energy intake, physical exercise level, smoking status and intensity of smoking, Deprivation Index, and Predictive Risk Modelling, including all the mean food groups (red and processed meat, fish, eggs, fibre-containing foods, nuts, fat, sweets and added sugar, and alcoholic beverage)	7
					Fresh cheese: T_3_ vs. T_1_	CRC: 0.92 (0.58, 1.46)		
					Other cheeses: T_3_ vs. T_1_	CRC: 1.87 (1.11, 3.16)		
Zhang et al., 2020 [33]	China (30–75)	2380 CRC (1476 CC, 828 RC, and 76 SCRC)	1356/102	2389 H	Total dairy products: T_3_ vs. T_1_	CRC: 0.32 (0.27, 0.39) CRC (M): 0.30 (0.23, 0.38) CRC (W): 0.36 (0.27, 0.47) CC: 0.31 (0.25, 0.38) RC: 0.35 (0.27, 0.45)	Age, sex, marital status, residence, educational level, occupation, income level, occupational activity, household and recreational PA, smoking status, alcohol drinking, family history of cancer, BMI, total energy intake, vegetable, fruit, red meat, and dietary fibre intake, and for W: age at menarche	7
					Milk, drink vs. not drink	CRC: 0.52 (0.45, 0.59) CRC (M): 0.49 (0.41, 0.59) CRC (W): 0.56 (0.46, 0.88) CC: 0.53 (0.46, 0.62) RC: 0.53 (0.44, 0.64)		
Collatuzzo et al., 2022 [34]	Iran (controls, 57.2+/−11.5; cases, 58.6+/−12.4)	865 CRC ^a^ (434 CC, 404 RC)	497/368	3205 C	Total dairy products: T_3_ vs. T_1_	CRC: 1.06 (0.85, 1.32) CC: 1.00 (0.75, 1.34) PCC: 0.98 (0.61, 1.58) DCC: 0.96 (0.62, 1.47) RC: 1.06 (0.78, 1.44)	Sex, age, BMI, smoking, opium, province, aspirin, SES, PA, use of red and processed meat, fat intake, fibre intake	5
					Yogurt: T_3_ vs. T_1_	CRC: 0.96 (0.77, 1.20) CC: 0.78 (0.58, 1.06) PCC: 0.43 (0.27, 0.70) DCC: 0.81 (0.52, 1.26) RC: 1.07 (0.80, 1.45)		
					Milk: T_3_ vs. T_1_	CRC: 0.98 (0.79, 1.21) CC: 1.06 (0.80, 1.41) PCC: 1.18 (0.74, 1.88) DCC: 1.30 (0.87, 1.96) RC: 0.97 (0.72, 1.31)		
					Dough: T_3_ vs. T_1_	CRC: 1.26 (0.98, 1.61) CC: 1.15 (0.83, 1.60) PCC: 1.52 (0.88, 2.61) DCC: 1.06 (0.65, 1.73) RC: 1.36 (0.96, 1.91)		
					Kashk: T_3_ vs. T_1_	CRC: 1.03 (0.81, 1.31) CC: 1.09 (0.79, 1.49) PCC: 0.90 (0.52, 1.58) DCC: 0.91 (0.57, 1.44) RC: 1.01 (0.73, 1.40)		
					Cheese: T_3_ vs. T_1_	CRC: 1.08 (0.81, 1.44) CC: 1.08 (0.74, 1.56) PCC: 0.78 (0.40, 1.49) DCC: 1.20 (0.70, 2.05) RC: 0.96 (0.63, 1.47)		
					Cream: T_3_ vs. T_1_	CRC: 1.33 (1.08, 1.64) CC: 1.37 (1.03, 1.81) PCC: 1.68 (1.08, 2.61) DCC: 0.93 (0.60, 1.43) RC: 1.20 (0.90, 1.60)		
					Ice cream: T_3_ vs. T_1_	CRC: 0.86 (0.62, 1.21) CC: 0.75 (0.48, 1.17) PCC: 1.48 (0.68, 3.22) DCC: 0.44 (0.23, 0.85) RC: 0.98 (0.61, 1.55)		
					Other milk products: T_3_ vs. T_1_	CRC: 1.00 (0.73, 1.37) CC: 0.99 (0.65, 1.50) PCC: 1.04 (0.53, 2.03) DCC: 1.07 (0.60, 1.92) RC: 0.96 (0.63, 1.47)		

BMI: body mass index; C: community controls; CC: colon cancer; CI: confidence interval; CRC: colorectal cancer; DCC: distal colon cancer; H: hospital controls; M: men; NOS: Newcastle-Ottawa Scale; PA: physical activity; PCC: proximal colon cancer; RC: rectal cancer; SES: socioeconomic status; SCRC: synchronous colon and rectal cancer; T: tertile; W: women. ^a^ Twenty-seven cases with unknown sites within the colorectum.

**Table 3 nutrients-14-03430-t003:** Summary of the findings found in the reviewed articles examining the association between the consumption of dairy products and the risk of colorectal cancer.

Food Type	Cohort Studies		Case-Control studies	
	No. of Total Studies (Ref.)	No. of Studies (Ref.), Type of Association, Food Subtype ^a^, CRC Overall or Subsites Risk ^b^	No. of Total Studies (Ref.)	No. of Studies (Ref.), Type of Association, Food Subtype ^a^, CRC Overall or Subsites Risk ^b^
Total dairy products in overall	5 [19,20,21,26,27]	3 [19,20,21], inverse, CRC	3 [32,33,34]	1 [33], inverse, CRC
Total dairy products by fat content	1 [19]			
Total milk in overall	5 [18,19,22,24,26]		2 [33,34]	1 [33], inverse, CRC
Total milk by fat content	1 [19]	1 [19], inverse, low-fat milk, CRC		
Yogurt and other fermented dairy products	4 [19,23,24,26]	1 [23], inverse, CC	1 [34]	
Cheese	4 [19,22,24,26]	1 [26], positive, “fromage blanc”, CRC	2 [32,34]	1 [32], positive, high-fat cheese, CRC
Other dairy products	3 [19,24,26]	1 [26], positive, sugary dairy products, CRC	1 [34]	1 [34], positive, cream, CRC, CC, PCC; inverse, ice cream, DCC

CC: colon cancer; CRC: colorectal cancer; DCC: distal colon cancer; PCC: proximal colon cancer; ^a^ If applicable; ^b^ The rest of the studies showed no significant effect.

**Table 4 nutrients-14-03430-t004:** Characteristics of the five cohort studies included in the systematic review examining the association between the consumption of fish, white meat, and eggs and the risk of colorectal cancer.

Study, Year (Ref.)	Study Cohort, Country (Age, y)	No. Participants (M/W)	No. Incident Cases (M/W)	Follow-Up Length, y	Exposure	HR (95%CI)	Adjustments to HR	NOS Quality Score
Aglago et al., 2020 [28]	EPIC, 10 European countries (cases, 57.3+/−7.9; controls, 51.2+/−9.95)	142,241/333,919	2719/3572	14.9	Total fish and shellfish: Q_5_ vs. Q_1_	CRC: 0.88 (0.80, 0.96) CC: 0.89 (0.79, 1.00) RC: 0.88 (0.75, 1.04) PCC: 0.93 (0.79, 1.11) DCC: 0.89 (0.75, 1.07)	Stratified by age, sex, and centre. Adjusted for BMI, height, PA, smoking, education, and intakes of energy, alcohol, red and processed meat, fibre, and dairy products	7
					Oily fish: Q_5_ vs. Q_1_	CRC: 0.90 (0.82, 0.98) CC: 0.89 (0.80, 0.99) RC: 0.91 (0.78, 1.06) PCC: 0.81 (0.70, 0.95) DCC: 1.03 (0.87, 1.21)		
					Non-oily fish: Q_5_ vs. Q_1_	CRC: 0.91 (0.83, 1.00) CC: 0.90 (0.80, 1.01) RC: 0.96 (0.82, 1.13) PCC: 0.95 (0.80, 1.12) DCC: 0.85 (0.71, 1.01)		
Bradbury et al., 2020 [22]	UK Biobank Cohort Study, UK (40–69)	219,329/256,252	2609	5.7	Total fish: ≥3 times/week vs. <once/week	CRC: 0.95 (0.80, 1.13)	Stratified by age, sex, geographical region, and SES. Adjusted for education, smoking status, waist circumference, height, PA, alcohol intake, family history of CRC, aspirin or ibuprofen use, use of vitamin D and folate supplements and for W: parity, menopause, OCA and HRT use	6
					Poultry: ≥2 times/week vs. never	CRC: 0.96 (0.79, 1.18)		
Knuppel et al., 2020 [29]	UK Biobank Cohort Study, UK (37–73)	218,498/256,498	28,955	6.9	Poultry: per 30 g/day	CRC: 1.02 (0.91, 1.14) CC: 1.01 (0.88, 1.15) RC: 1.02 (0.85, 1.24) RC (M): 1.27 (1.00, 1.62) RC (W): 0.72 (0.52, 0.98)	Stratified for sex, age group, region; and adjusted for age, ethnicity, deprivation, qualification, employment, living with a spouse or partner, height, smoking, PA, alcohol intake, total fruit and vegetable intake, estimated cereal fibre intake, BMI, and for W: menopausal status, parity, HRT and OCA use	7
Mejborn et al., 2021 [30]	The Danish National Survey on Diet and Physical Activity cohort study, Denmark (>50)	3033/3249	127 CRC	8.7	Poultry: ≥16 g/day vs. <16 g/day	CRC: 1.62 (1.13, 2.31)	Sex, educational attainment, ethnicity, smoking, PA, alcohol, BMI, and total energy intake	7
Wang et al., 2022 [31]	NHS, NHSII, and HPFS, USA (M, 40–75; W, 25–42)	527/4742	404 CC, 122 RC	30	Total fish: 1 SD, MPS Oily fish: 1 SD, MPS Canned tuna fish: 1 SD, MPS	CRC: OR, 0.86 (0.78, 0.96) CRC: OR, 0.86 (0.77, 0.96) CRC: OR, 0.87 (0.78, 0.97)	BMI, family history of CRC, endoscopy, multivitamin use, aspirin use, smoking, PA, total energy intake, alcohol intake, and modified AHEI (in NHS/HPFS)	6
					Poultry: 1 SD, MPS	CRC: OR, 0.94 (0.85, 1.05)		

AHEI: alternate healthy eating index; BMI: body mass index; CC: colon cancer; CRC: colorectal cancer; DCC: distal colon cancer; EPIC: European Prospective Investigation into Cancer and Nutrition; HPFS: Health Professional Follow-up Study; HR: hazard ratio; HRT: hormone replacement therapy; M: men; MPS: metabolic profile score; NHS: Nurses’ Health Study; NOS: Newcastle–Ottawa Scale; OCA: oral contraceptive agent; OR: odds ratio; PA: physical activity; PCC: proximal colon cancer; Q: quintile; RC: rectal cancer; SD: standard deviation; SES: socioeconomic status; W: women.

**Table 5 nutrients-14-03430-t005:** Characteristics of the five case–control studies included in the systematic review examining the association between the consumption of fish, white meat, and eggs and the risk of colorectal cancer.

Study, Year (Ref.)	Country (Age, y)	No. Cases and Endpoint	Sex, No. of Cases (M/W)	No. Controls and Type	Exposure	OR (95% CI)	Adjustments to OR	NOS Quality Score
Deoula et al., 2019 [35]	Morocco (≥18)	1453 CRC	716/737	1453 C	White meat: >308 g/week vs. ≤308 g/week	CRC: 1.07 (0.96, 1.19) CRC (M): 1.08 (0.92, 1.26) CRC (W): 1.15 (0.93, 1.42) CC: 1.13 (0.97, 1.31) CC (M): 1.13 (0.91, 1.41) CC (W): 1.01 (0.80, 1.26) RC: 1.01 (0.86, 1.18) RC (M): 1.03 (0.82, 1.28) RC (W): 1.08 (0.92, 1.25)	Age, residence, education level, monthly income, PA intensity, smoking status, BMI, NSAIDs, total energy intake, calcium, dietary fibre, family history of CRC, dairy product, fruits, vegetable, fish, and alcohol consumption	6
					Turkey: >51 g/week vs. ≤51 g/week	CRC: 0.89 (0.80, 1.01) CRC (M): 0.94 (0.75, 1.18) CC: 0.92 (0.79, 1.08) CC (M): 0.90 (0.72, 1.13) RC: 0.85 (0.72, 1.01) RC (M): 0.92 (0.79, 1.08)		
					Poultry: >196 g/week vs. ≤196 g/week	CRC: 1.10 (0.99, 1.23) CRC (M): 1.15 (0.98, 1.35) CRC (W): 1.08 (0.92, 1.26) CC: 1.15 (0.98, 1.34) CC (M): 1.27 (1.01, 1.59) CC (W): 1.10 (0.88, 1.37) RC: 1.05 (0.89, 1.22) RC (M): 1.05 (0.84, 1.31) RC (W): 1.08 (0.86, 1.36)		
Kim et al., 2019 [36]	Republic of Korea (cases, 56.6+/−9.7; controls, 56.1+/−9.1)	923 CRC	625/298	1846 C	Total fish and shellfish: T_3_ vs. T_1_	CRC: 1.04 (0.82, 1.32) CRC (M): 0.99 (0.74, 1.32) CRC (W): 1.25 (0.81, 1.94)	Total energy intake, BMI, first-degree family history of CRC, occupation, educational level, monthly income, marital status, regular exercise, and alcohol consumption	6
Alegria-Lertxundi et al., 2020 [32]	Spain (50–69)	308 CRC (234 DCC, 74 PCC)	204/104	308 C	Eggs: T_3_ vs. T_1_	CRC: 1.26 (0.71, 2.23)	Age, sex, BMI, energy intake, physical exercise level, smoking status and intensity of smoking, Deprivation Index, and Predictive Risk Modelling, including all the mean food groups (red and processed meat, fibre-containing foods, nuts, fat, sweets and added sugar, and alcoholic beverage)	7
					Total fish: T_3_ vs. T_1_ Oily fish: T_3_ vs. T_1_ Non-oily fish: T_3_ vs. T_1_	CRC: 1.25 (0.68, 2.29) CRC: 0.53 (0.27, 0.99) CRC: 1.29 (0.74–2.25)		
Shen et al., 2021 [37]	China (cases, 60.3+/−13.4; controls, 59.6+/−12.9)	100 CRC	54/46	100 C	Eggs: ≥280 g/week vs. <280 g/week	CRC: 0.26 (0.10, 0.69)	Age, BMI	4
					White meat: >500 g/week vs. ≤500 g/week	CRC: 0.86 (0.30, 2.46)		
Franchi et al., 2022 [38]	Italy (50–69)	2419 CRC (727 DCC, 373 PCC)	1432/987	4723 H	Total fish (canned and non-canned) vs. no fish	CRC: 0.69 (0.58, 0.81)	Centre, study, sex, age, BMI, education, family history of CRC, PA at work, smoking habits, alcohol consumption, vegetable and fruit consumption, and energy intake	5
					Non-canned fish vs. no fish	CRC: 0.88 (0.77, 1.00)		
					Only canned fish vs. no fish	CRC: 0.77 (0.62, 0.97)		
					Canned fish: ≥2 servings/week vs. <1 serving/week	CRC: 0.86 (0.51, 0.85) CC: 0.66 (0.49, 0.90) RC: 0.65 (0.44, 0.95)	The same as in the previous row plus fish consumption	

BMI: body mass index; C: community controls; CC: colon cancer; CI: confidence interval; CRC: colorectal cancer; DCC: distal colon cancer; H: hospital controls; M: men; NOS: Newcastle–Ottawa Scale; NSAIDs: non-steroidal anti-inflammatory drugs; OR: odds ratio; PA: physical activity; PCC: proximal colon cancer; R: rectal cancer; T: tertile; W: women.

**Table 6 nutrients-14-03430-t006:** Summary of the findings found in the reviewed articles examining the association between the consumption of fish, white meat, and eggs and the risk of colorectal cancer.

Food Type	Cohort Studies		Case-Control Studies	
	No. of Total Studies (Ref.)	No. of Studies (ref.), Type of Association, CRC Overall or Subsites Risk ^a^	No. of Total Studies (Ref.)	No. of Studies (Ref.), Type of Association, CRC Overall or Subsites Risk ^a^
Total fish	3 [22,28,31]	2 [28,31], inverse, CRC and CC	3 [32,36,38]	1 [38], inverse, CRC
Oily fish	2 [28,31]	2 [28,31], inverse, CRC, CC, and PCC	1 [32]	1 [32], inverse, CRC
Non-oily fish	1 [28]	1 [28], inverse, CRC, CC, and DCC	1 [32]	
Canned fish	1 [31]	1 [31], inverse, CRC	1 [38]	1 [38], inverse, CRC
White meat	1 [37]		1 [35]	
Poultry	4 [22,29,30,31]	1 [29], inverse, RC (W); positive, RC (M)	1 [35]	1 [35], positive, CC (M)
Turkey			1 [35]	
Egg			2 [32,37]	1 [37], inverse, CRC

CC: colon cancer; CRC: colorectal cancer; DCC: distal colon cancer; M: men; PCC: proximal colon cancer; RC: rectal cancer; W: women. ^a^ The rest of the studies showed no significant effects.

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
