# Peer review of "Role of Dairy Foods, Fish, White Meat, and Eggs in the Prevention of Colorectal Cancer: A Systematic Review of Observational Studies in 2018–2022"

_nutrients, 2022, doi:10.3390/nu14163430_

Round 1

Reviewer 1 Report

This systematic review shows that a high consumption of total dairy products was associated with a lower colorectal cancer risk. It is an interesting results, however I can not understand what the new findings are.

Major comments

1.      You should clearly show not only significant results but also new findings.

2.      Line 226-227, in studies for Chinese people, it is possible that people who consume a lot of dairy products are also consuming more meat (red meat and processed meat) because their diet is changing from traditional diet to western diet. You should consider this point.

3.      You described “this review on white meat and egg was based on a small number of studies and the evidence was low, thus findings for these food groups should be interpreted with caution” in Conclusion. You should describe so in Abstract.

Author Response

Dear Reviewer #1,

Thank you for taking the time to provide thoughtful, detailed feedback. We are greatly appreciative of your time and have made the requested revisions. We believe our revised paper is stronger than previous iterations and we are excited for your second review. Below you will find your comments and our responses. We reference sections and line numbers for easier readership.

Thank you for your time and consideration,

       Authors

Reviewer' Comments to Author:

1. You should clearly show not only significant results but also new findings.

We included additional information in the Conclusions section, lines 1254-69: “The current findings related to dairy products and fish confirm those from previous meta-analyses [7,12,67,68]. The main new findings of the present review are those related to specific subtypes of fish (in particular, canned fish), white meat and egg”.

2. Line 226-227, in studies for Chinese people, it is possible that people who consume a lot of dairy products are also consuming more meat (red meat and processed meat) because their diet is changing from traditional diet to western diet. You should consider this point.

We introduced the following sentences in the manuscript, subsection 3.3.1, lines 449-51: “(for example, in China, generally, the levels of dairy consumption are low, but people who has a high consumption of dairy products can also have a high intake of red and processed meat, due to the westernization of eating habits).”

3. You described “this review on white meat and egg was based on a small number of studies and the evidence was low, thus findings for these food groups should be interpreted with caution” in Conclusion. You should describe so in Abstract.

We appreciate the reviewer's comment. We introduced the following sentence in the manuscript: Abstract, lines 23-24: “The association between white meat and egg intake and CRC risk was low, and based on a small number of studies, thus these findings should be interpreted with caution.”

Reviewer 2 Report

The opinion of this reviewer is present manuscript is well structured and complete in all sections.

However, there are some suggestions that I feel I can give to the authors and that would give, in my opinion, greater completeness and scientific impact to this review. These suggestions include slight modifications in introduction and discussion sections.

In fact, some hints to the benefits deriving from the consumption of animal-source foods or from the adoption of a Mediterranean diet, as recently reported in other studies and reviews.

Just citing few example, in line 47 it would be appropriate to cite benefits connected to eggs consumption as demonstrated by Pujia R. and her collaborators (1), as well as those connected to Mediterranean diet as described by Mazza and coworkers (2).

Similarly, I would have briefly cited and mentioned studies describing CRC innovative treatments deriving from natural and biocompatible extracts, such as those deriving from grape seeds (3) or Salvia spp vegetables (4), to cite a few.

In fact, in the opinion of this reviewer, treating this topic, perhaps within few lines, would allow you to better complete and structure an already well-conceived review, obviously being careful not to be misleading or move away from the target topic of the review itself.

1- Zaheer, K. An updated review on chicken eggs: production, consumption, management aspects and nutritional benefits to human health. Food and Nutrition Sciences 2015, 6, 1208.

2 - E. Mazza et al. - Mediterranean Diet In Healthy Aging. - J Nutr Health Aging. 2021;25(9):1076-1083. doi:10.1007/s12603-021-1675-6.

3 - M. Iannone et al. - Characterization and in vitro anticancer properties of chitosan-microencapsulated flavan-3-ols-rich grape seed extracts. - Int J Biol Macromol. 2017 Nov;104(Pt A):1039-1045. doi:10.1016/j.ijbiomac.2017.07.022. Epub 2017 Jul 4.

4 - Vila, L.; Marcos, R.; Hernández, A. Long-term effects of silver nanoparticles in caco-2 cells. Nanotoxicology 2017, 11, 771-780.

Author Response

Dear Reviewer #2,

Thank you for taking the time to provide thoughtful, detailed feedback. We are greatly appreciative of your time and have made the requested revisions. We believe our revised paper is stronger than previous iterations and we are excited for your second review. Below you will find your comments and our responses. We reference sections and line numbers for easier readership.

Thank you for your time and consideration,

       Authors

Reviewer' Comments to Author:

The opinion of this reviewer is present manuscript is well structured and complete in all sections.

However, there are some suggestions that I feel I can give to the authors and that would give, in my opinion, greater completeness and scientific impact to this review. These suggestions include slight modifications in introduction and discussion sections.

In fact, some hints to the benefits deriving from the consumption of animal-source foods or from the adoption of a Mediterranean diet, as recently reported in other studies and reviews.

Just citing few example, in line 47 it would be appropriate to cite benefits connected to eggs consumption as demonstrated by Pujia R. and her collaborators (1), as well as those connected to Mediterranean diet as described by Mazza and coworkers (2).

Similarly, I would have briefly cited and mentioned studies describing CRC innovative treatments deriving from natural and biocompatible extracts, such as those deriving from grape seeds (3) or Salvia spp vegetables (4), to cite a few.

In fact, in the opinion of this reviewer, treating this topic, perhaps within few lines, would allow you to better complete and structure an already well-conceived review, obviously being careful not to be misleading or move away from the target topic of the review itself.

1- Zaheer, K. An updated review on chicken eggs: production, consumption, management aspects and nutritional benefits to human health. Food and Nutrition Sciences 20156, 1208.

2 - E. Mazza et al. - Mediterranean Diet In Healthy Aging. - J Nutr Health Aging. 2021;25(9):1076-1083. doi:10.1007/s12603-021-1675-6.

3 - M. Iannone et al. - Characterization and in vitro anticancer properties of chitosan-microencapsulated flavan-3-ols-rich grape seed extracts. - Int J Biol Macromol. 2017 Nov;104(Pt A):1039-1045. doi:10.1016/j.ijbiomac.2017.07.022. Epub 2017 Jul 4.

4 - Vila, L.; Marcos, R.; Hernández, A. Long-term effects of silver nanoparticles in caco-2 cells. Nanotoxicology 201711, 771-780.

We appreciate the reviewer's comment. We introduced the following sentences in the manuscript:

Introduction, lines 43-53: “In addition, Mediterranean dietary pattern could reduce overall cancer risk [8] and, in particular, CRC risk [9]. On the other hand, recently, CRC innovative treatments deriving from natural extracts, termed nutraceuticals, have gained interest [10]. In this sense, for example, anti-inflammatory and reparative properties have been attributed to the nutraceutical grape seed extract [11].”

Subsection 3.5.3, lines 1221-34: “Although Shen et al. [37] did not provide arguments to explain why egg consumption is a protective factor against CRC risk, it should be noted that eggs contain several antioxidants (including selenium, carotenoids, and vitamin E) which reduce free radicals arising from cellular metabolism [76]. Positive associations between oxidative stress and the incidence of chronic diseases such as cancers have been reported [77]. In addition, eggs are a good source of choline and B vitamins directly or tangentially involved in one-carbon metabolism [76]. Disruption of one-carbon metabolism can lead to decreased DNA synthesis, genomic instability, and decreased methyl donor production [78]. CRC is associated with genomic instability and DNA hypotethylation [79]. On the contrary, in the meta-analysis of Schwingshackl et al. [14], based on a small number of studies (n=3), results showed a positive significant association in the high vs. low meta-analysis. A plausible hypothesis to explain these observations is that a high-cholesterol intake would increase the formation of secondary bile acid and enhance the induction of colorectal tumours [80].”

Round 2

Reviewer 1 Report

Manuscript is well revised, according to suggestions.

Author Response

Dear Reviewer,

Thank you for your time and consideration,

   Authors